# Methylglyoxal in the Brain: From Glycolytic Metabolite to Signalling Molecule

**DOI:** 10.3390/molecules27227905

**Published:** 2022-11-15

**Authors:** Zeyong Yang, Wangping Zhang, Han Lu, Shu Cai

**Affiliations:** 1Department of Anesthesiology, International Peace Maternity and Child Health Hospital, Shanghai Jiao Tong University School of Medicine, Shanghai Key Laboratory of Embryo Original Disease, Shanghai Municipal Key Clinical Specialty, Huashan Rd. 1961, Shanghai 200030, China; 2Department of Anesthesiology, Women and Children’s Hospital of Jiaxing University, No. 2468 Zhonghuan East Road, Jiaxing 314000, China; 3Department of Anesthesiology, Ruijin Hospital, Shanghai Jiao Tong University School of Medicine, Shanghai 200025, China; 4School of Nursing, Guangdong Pharmaceutical University, No. 283 Jianghai Avenue, Haizhu District, Guangzhou 510310, China

**Keywords:** methylglyoxal, glyoxalase, neurodegenerative, bioenergetics, side-product, homeostasis, behavioural phenotypes

## Abstract

**Highlights:**

**Abstract:**

Advances in molecular biology technology have piqued tremendous interest in glycometabolism and bioenergetics in homeostasis and neural development linked to ageing and age-related diseases. Methylglyoxal (MGO) is a by-product of glycolysis, and it can covalently modify proteins, nucleic acids, and lipids, leading to cell growth inhibition and, eventually, cell death. MGO can alter intracellular calcium homeostasis, which is a major cell-permeant precursor to advanced glycation end-products (AGEs). As side-products or signalling molecules, MGO is involved in several pathologies, including neurodevelopmental disorders, ageing, and neurodegenerative diseases. In this review, we demonstrate that MGO (the metabolic side-product of glycolysis), the GLO system, and their analogous relationship with behavioural phenotypes, epigenetics, ageing, pain, and CNS degeneration. Furthermore, we summarise several therapeutic approaches that target MGO and the glyoxalase (GLO) system in neurodegenerative diseases.

## 1. Methylglyoxal (A Metabolic Side-Product) and the GLO System In Vivo

Methylglyoxal (MGO) is an unavoidable endogenous coproduct of the metabolism of carbohydrates, lipids, and proteins that is produced either autonomously or enzymatically [1] primarily via nonenzymatic degradation of the triphosphate intermediates of glycolysis. 

The glyoxalase (GLO) system primarily catalyses the detoxification of MGO and other reactive aldehydes. In the glutathione (GSH)-dependent GLO1 system, MGO is transformed into S-D-lactoylglutathione and then into D-lactate via the GLO2 system. Concurrently, GLO1 activity limits the reaction rate of MGO degradation and regulates MGO-induced toxicity. Many of the cellular signalling pathways that regulate GLO1 prevent the formation of advanced glycation end-products (AGEs) during MGO anabolism or catabolism [2]. 

It is of paramount importance to thoroughly study antiapoptotic GLO2 overexpression [3]. Transduced recombinant Tat-GLO1, Tat-GLO2, or both have been studied, and these are believed to defend HT22 cells against the toxicity of MGO and H_2_O Indeed, Tat-GLO1 and Tat-GLO2 have been shown to provide toxicity protection in vivo in diabetic mouse models with ischaemic damage [4]. Interestingly, a substantial increase in intracellular Tat-GLO2 has a protective effect on HT22 cells but to a lesser degree than a similar increase in Tat-GLO The activities of GLO1 or GLO2 can increase pyruvaldehyde elimination. Under normal conditions, cells are not affected by MGO-induced toxicity, and the GLO system is the most pivotal signal pathway for the detoxication of this substance.

However, more experimental strategies are necessary. Liquid chromatography–mass spectrometry (LC-MS/MS) of glycation adducts is an excellent technique for comparing various AGE sources; however, this method does not provide relevant data on homologous proteins. Nonetheless, because both the formation and degradation of glycation products depend on the protein sequence and structure to some extent [5], this information is needed to study functional changes in glycation-related proteins; with proteomics, we can directly identify and quantify a single glycation site. GLO1 and GLO2 activities have been shown to be impaired in some circumstances [6], and it is likely that glutathione is involved in phytochelatin biosynthesis.

The cerebral GLO system can adapt adequately to eliminate MGO-induced hazardous substances such as side-products while still supplying sufficient energy for brain functioning (Figure 1).

## 2. Potential Role of MGO in Ageing via Its Induction of Oxidative Stress, Neuropathic Pain, an Anxiolytic Effect, and Apoptosis [7]

Estimates of MGO in mouse brains derived using our approach were 3 and 5 μM in the brain cortex and the midbrain and brain stem combined, respectively, which matches the predicted values [8]. High MGO levels can render neurons susceptible to AGE formation, a risk factor linked to the development of neurodegenerative disorders [9]. MGO can induce oxidative stress and cause irreversible loss of protein function, including protein cross-linking.

MGO accumulation has been implicated in multiple neurodegenerative diseases, and AGEs in combination with MGO can potentially induce ageing and diabetes-associated complications. Occurrences of these metabolic adaptations of neural cells have been found in *Drosophila* [10] and mice [11]. The key role of glycolytic-derived lactate released from astrocytes during neural activities has been identified [12]: this lactate acts as an intercellular chemical messenger [13] and metabolic fuel [14]. By observing the brain mitochondria bioenergetics and oxidative status of rats that have undergone chronic treatment with MGO and/or pyridoxamine (a glycation inhibitor), it has been found that the resulting increased MGO levels induce mitochondrial impairment in the central nervous system (CNS) and that pyridoxamine cannot reverse MGO-mediated effects.

Increased intracellular MGO may induce cytotoxicity in INS-1 cells. This occurs primarily via the activation of oxidative stress and is further amplified by its triggering of the mitochondrial apoptotic pathway and the endoplasmic reticulum (ER) stress-evoked JNK pathway, which are involved in glucotoxicity-mediated pancreatic beta-cell apoptosis. Furthermore, the cytotoxicity of MGO is mediated via the regulation of the amount of deoxyribonucleic acid and the activation of apoptosis. In the development of diabetic complications, MGO can modify cellular proteins (i.e., cross-linking amino groups), which then generate AGEs. Numerous studies have highlighted the causal relationship between MGO-derived AGEs and diabetic complications and the role of AGEs in endothelial dysfunction [15]. 

It has been demonstrated that AGEs can induce corneal endothelial cell apoptosis by increasing cellular oxidative stress [16]. MGO-induced reactive oxygen species (ROS) and AGEs can impair mitochondrial function, resulting in further ROS production and damage. AGE accumulation increases significantly after MGO treatment and can be alleviated via treatment with Tan IIA. Furthermore, Tan IIA treatment also attenuates the production of ROS, TBARS, and H_2_O_2_ in cultured human brain microvessel endothelial cells (HBMECs). These findings indicate that Tan IIA can protect HBMECs by inhibiting AGE accumulation and reducing oxidative stress.

In addition, MGO requires between several hours and days to induce apoptosis. Furthermore, the effects that MGO has on GABAA receptors occur only at lower concentrations and are momentarily dissociated because of their more prominent effects, such as cytotoxicity and AGE formation. Diabetes, which has been linked to stroke, can elevate blood MGO levels via an increase in MGO secretion due to high glucose levels, thus impairing the GSH-GLO system, which accelerates MGO elimination. MGO also glycates proteins and causes dicarbonyl stress. This is evident in the diabetic brain, in which there is only a tiny chance of GSH-dependent MGO elimination, resulting in elevated protein glycation and oxidative or carbonyl stress. Furthermore, these MGO-induced changes may exacerbate post-stroke brain damage. Administering N-acetylcysteine (NAC) may alleviate post-stroke brain injury damage by restoring GSH generation and normalising the MGO-to-GSH ratio, thus diminishing oxidative or carbonyl stress levels. This treatment may be a critical factor in the management of diabetic stroke risk and outcomes. 

There may be a metabolic imbalance following acute neuronal lesions. An increase in the rate of glycolysis is accompanied by MGO formation, leading to metabolic dysfunction and inflammation. As mentioned earlier, the GLO system is the primary MGO detoxification system; however, it can be impaired following excitotoxicity and stroke. Changes in GLO1 protein content are associated with excitotoxicity but seemingly not with fibre transection. Cell-specific changes in GLO1 immunoreactivity following different in vitro and in vivo lesion types may be a common phenomenon in the aftermath of neuronal lesions [15]. Furthermore, MGO triggers protein and nucleotide modification (e.g., AGEs), ROS production, and apoptosis. MGO has been shown to increase the production of ROS in other animal disease models [16]. In addition to directly increasing ROS production, MGO can also increase oxidative stress by inducing AGE formation. Specifically, MGO modifies proteins and DNA to form hydroimidazolone MGO-H1 and imidazopurinone MGOdG adducts, respectively. An abnormal accumulation of MGO and dicarbonyl stress increases adduct levels, which may induce apoptosis and replication catastrophe [17]. 

There is mounting evidence supporting the role of GLO1 and its substrate (MGO) in the regulation of anxiety-like behaviour [18,19,20]. Changes in the brain expression levels of GLO1 and Gsr produce a significant impact on anxiety-related behaviours, thus establishing the causal role of these genes. GLO1 plays an essential role in MGO clearance via the excessive secretion of GLO1, preventing MGO accumulation and inhibition of GLO1, resulting in MGO aggregation [18,19,20]. Evidence has been presented showing that MGO can reduce anxiety-like behaviour in an elevated plus maze. The research findings referenced here are based on the results of administering MGO to the basolateral amygdala (BLA), which is closely linked with anxiety-like behaviour [21,22,23,24]. Furthermore, astrocytes and neurons have different energy-adaptive mechanisms for GLO defence mechanisms, which may function as a shield mechanism, preventing MGO from causing cellular injury. MGO increases intracellular calcium in sensory neurons and causes behavioural injury via the transient receptor potential ankyrin 1 (TRPA1) channel. In addition, after the administration of MGO in wild-type rats and mice, the immunohistochemical phosphorylation of extracellular signal-regulated kinase (p-ERK) and multiple pain-like behaviours were evaluated. It was observed that MGO stimulated certain behaviours, including conditioned place avoidance (a measure of affective pain), dose-dependent licking, enhancement of injury perception, heat hypersensitivity, and mechanical stimulation.

MGO-mediated dicarbonyl adducts have complex pleiotropic effects on biological processes in cells. These adducts can modulate protein activity and stability and generate ROS and oxidative stress [25], which may induce distinct outcomes [25]. They have been shown to contribute to oxidative DNA damage and apoptosis. DNA glycation is associated with elevated mutation frequency, DNA strand breaks, and cytotoxicity [26]. Parkinsonism-associated protein DJ-1 is a major nucleotide repair system. Furthermore, some MGO-derived AGEs possess antioxidant properties [27].

GLO1- and MGO-derived AGEs play major roles in vascular physiology and pathophysiology, including the etiopathogenesis of brain microvascular endothelial barrier dysfunctions [28,29]. High concentration levels of MGO can induce mitochondrial injury, ROS generation, and oxidative-stress-mediated cell damage [30]. In addition, MGO toxicity within cells or tissues is mediated primarily via an enhancement of oxidative stress and apoptosis abduction. In light of these processes, oxidative stress is considered to play a crucial role in the pathogenesis of Alzheimer’s disease (AD) [31,32]. A marked increase (less than three-fold) took place in AMPK phosphorylation at 3, 6, and 18 h after 5 mM MGO treatment, indicating that autophagy was induced through AMPK activation. This strong AMPK activation was confirmed by enhanced phosphorylation of the AMPK substrate. AMPK phosphorylates and activates the tuberous sclerosis complex and then phosphorylates the small GTPase Ras homolog enriched in the brain. This is done to prevent mTOR activation. DNA glycation is linked to elevated mutation frequency, DNA strand breaks, and cytotoxicity. Parkinsonism-associated protein DJ-1 is a major nucleotide repair system [26] (Figure 2).

## 3. Toxicity of MGO and Some Diseases (e.g., CNS Degenerative Diseases, Pain, Hypertension, Ageing, Epigenetics, Diabetes-Related Neurological Complications, and Anxiety-Related Behaviour)

Recently, there have been many findings linking GLO1 to CNS degenerative diseases, hypertension, ageing, epigenetics, diabetes-related neurological complications, and several behavioural phenotypes (Figure 3). There has been an emphasis on discussing the toxicity of MGO and some diseases, such as CNS degenerative diseases, pain, atherosclerosis, hypertension, ageing, epigenetics, diabetes, and anxiety-related behaviour. 

### 3.1. MGO-GLO System and CNS Degeneration 

MGO is a major precursor and a reactive dicarbonyl intermediate of AGEs. Sugars are produced from metabolised glucose and then react covalently with proteins to produce AGEs, which can impact protein function [23]. Alzheimer’s patients present elevated levels of MGO in their cerebrospinal fluid and elevated AGE formation [33,34]. Cognitive decline has also been linked to disturbances in MGO metabolism that occur with ageing [35]. Pathologically, high brain levels of reactive dicarbonyls, such as MGO or glyoxal, initiate processes that lead to neurodegeneration, which presents clinically as cognitive or motor impairment disorders. Oxidative stress causes many pathological changes that lead to the vascular dangers of neurodegenerative diseases and ageing. The toxicity of MGO to tissues or cells develops via an increase in oxidative stress and induced apoptosis. In light of these processes, oxidative stress is considered a key feature in the developmental process of AD [31]. 

The regulation of GLO1 activity is closely connected to MGO toxicity levels because it is the rate-limiting step in MGO degradation [36,37]. Studies on the toxicity of MGO have revealed that thiols are MGO targets responsible for decreasing GSH levels. Because of the high reactivity of MGO, it is unclear whether a spontaneous reaction with GSH prevents MGO from reacting with targets that are more sensitive than itself, thus averting permanent enzyme and DNA damage. Reactive thiols are sensitive targets of MGO toxicity. For instance, it is believed that the MGO-dependent post-translational modification of cysteine residues in the TRPA1 pain receptor opens the TRPA1 channel [38]. In this regard, the effects of MGO on the reactive cysteines of thiolic proteins may potentially be a mechanism underpinning the toxic effects of MGO.

Relevant research findings indicate that, while serving a key role in the pathogenesis of various neurodegenerative disorders, MGO can cause cell damage, protein cross-linking, and glycation. MGO increases the production rate of amyloids, oligomers, and protofibrils and enhances the dimensions of the aggregates [39]. Apolipoprotein E (ApoE) is a vital gene susceptible to neural apoptosis and neural diseases [40,41]. Compared with noncarriers of ApoE4, patients who are ApoE4 carriers present a stronger Aβ secretion in the form of senile plaques when ApoE is deposited [42,43]. MGO is primarily produced via glycolysis, and when neurons are compared with astrocytes, the ability of neurons to increase glycolytic flux is poorer than that of astrocytes, which may be attributed to weak defence mechanisms against MGO-dependent toxicity. Consequently, the neuroenergetic specialisation of cells may function as a protective mechanism against neurotoxicity produced by MGO. MGO accumulation is detrimental because this metabolic substance is a potent glycating agent in cells. It is prone to reacting with lipids, nucleic acids, lysine proteins, and arginine residues to produce AGEs, including hydroimidazolone MGO-H1, argpyrimidine, *N*-(1-carboxyethyl)lysine (CEL), and MGO-derived lysine dimers [44].

AGEs are associated with a variety of pathophysiological mechanisms, such as diabetes complications, ageing, neural apoptosis, and neuro-disorders [45,46]. Impaired GLO1 activity is similarly capable of inducing elements of the pathogenesis of neurodegenerative disorders, including induced oxidative stress and AGE formation. MGO is associated with numerous age-related pathologies. For example, in patients with AD, GLO1 levels are elevated during the early phases of the disease but decline during the later stages due to an apparent increase in AGE accumulation [47,48]. 

In AD models, the most remarkable characteristic of AGE colocalisation is accompanied by amyloid plaques and neurons with hyperphosphorylated tau proteins. In addition, astrocytes can be found with AGE elevation and an inducible nitric oxide synthase immune response [45]. Extensive protein cross-linking and oxidative stress are also associated with glycation in AD, and glycation contributes to the production of AGEs. Similar differential responses have been observed, showing that MGO and MGO-derived AGEs play a key role in the etiopathogenesis of AD.

There has been an emphasis on Parkinson’s disease (PD), a long-term degenerative disorder of the CNS. Diabetes can increase the risk of PD. Hyperglycaemia has a positive correlation with MGO levels, which may be a critical factor in the molecular construction of PD and other synucleinopathies; this correlation may be via regulating the aggregation, accumulation, and neurotoxicity of α-synuclein. It is generally known that glycation and PD are genetically linked. This link stems from the mutations that occur in DJ-1 *PARK7*, a protein known as an anti-MGO enzyme, which undergoes GLO and deglycase activities [47,49] associated with recessive forms of PD. Interestingly, the activities of several glycated proteins, such as glyceraldehyde-3-phosphate dehydrogenase, aldolase, and aspartate amino transferase, are restored by DJ-Excessive protein glycation in PD cases associated with DJ-1 mutations may lead to abnormal protein secretion. In line with these characteristics, physical interactions between DJ-1 and α-synuclein have been identified: human wild-type DJ-1 and yeast DJ-1 orthologues can defend against the toxicity associated with α-synuclein. In addition, antiglycation enzymes (DJ-1, GLO1, or MGO) can lead to novel and intriguing therapeutic strategies for synucleinopathies. Glycation may constitute an important link in the molecular pathogenesis of PD and other synucleinopathies, which may be a second hit that enhances the risk of developing a disorder in a large portion of synucleinopathy patients [50]. It is worth noting that Glo4, another component of the GLO system, emerged as an α-synuclein side effect enhancer on a genetic screen in yeast [26].

### 3.2. MGO and Diabetes-Relevant Neurological Complications 

MGO is a kind of highly active α-oxoaldehyde, which is primarily formed from the triose phosphate intermediates of glycolysis, dihydroxyacetone phosphate, and glyceraldehyde 3-phosphate. As the serum levels of MGO within diabetic patients increase, MGO becomes more involved with diabetic complications, including cognitive impairment [51]. Type 2 diabetes is a metabolic disorder characterised by elevated blood glucose levels. The chronic hyperglycaemic state of diabetes contributes to most vascular complications of the disease, and the mechanism most commonly proposed as underpinning this state is glycating chemistry mediated by MGO, which accumulates in patients with type 2 diabetes. The ability of MGO to diffuse its production and react with α-synuclein can be used as a theoretical basis to explain the elevated incidence of PD in patients with type 2 diabetes [52]. Specifically, neural precursors in the developing murine cortex are regulated by a perturbed GLO1-MGO pathway in patients with gestational diabetes [53].

### 3.3. MGO and Inflammation

There is considerable evidence indicating that elevated MGO levels can stimulate an inflammatory reaction, activate endothelial cells, and induce cell dysfunction and vascular damage. MGO can also aggregate and induce inflammatory reactions and vascular injury in individuals with diabetes. Various studies have reported that MGO in individuals with diabetes enhances inflammatory responses, leading to endothelial cell loss. Although glucose is the primary energy substrate of the brain, hyperglycaemia aggravates ischaemic brain damage in acute stroke. Hyperglycaemia commonly occurs in patients suffering from acute stroke and typically denotes a poor outcome. Nonetheless, the relevant mechanisms that underpin the detrimental effects of hyperglycaemia need to be explored further. Previously obtained evidence shows that in the brain of ischaemic mouse models, hyperglycaemia can lead to an increase in infarct size and a decrease in the amount of protective noninflammatory monocytes/macrophages. Endothelial CSF-1 (M-CSF) stimulates the polarisation of noninflammatory macrophages in the ischaemic brain [54]. Importantly, as a novel strategy for treating stroke patients, deleting AGEs can normalise monocyte or macrophage polarisation and reverse the detrimental effects of hyperglycaemia [54].

A physiological decline in GLO1 activity and expression occurs with ageing, as demonstrated by Morcos et al. in a study on the nematode *Caenorhabditis elegans*, in which an inverse correlation was found between ageing and GLO1 expression [55]. This same effect has been confirmed in rodents [56,57,58]. Therefore, GLO1 expression is likely linked to healthy ageing [59].

In addition, the development and clinical outcome of vascular complications include endothelial disorders, such as retinal disease, impaired wound healing, and large vascular disease. Hydrolysing of AGE-modified proteins produces AGE-free adducts or glycated amino acids, which can be removed from the body by the urinary system. Dietary AGEs mainly supplement the endogenous flux in AGE-free adduct formation. A key precursor of AGEs is the dicarbonyl metabolite, MGO, which is metabolised by GLO1 in the cytoplasmic GLO system.

### 3.4. MGO and Anxiety-Related Behaviour

GLO1, a cytoplasmic enzyme, catalyses the reaction of glutathione with acyclic a-oxalaldehyde, especially MGO [21]. GLO1 increases anxiety levels by reducing the GABAA receptor agonist MGO [19], which makes GLO1 a potential target for anxiety disorders. Indeed, GLO1 duplication was bred into *Fkbp5* (FK506-binding protein 5) knockout mice, and it had no effect on anxiety [60]. Some initial studies have claimed that GLO1 gene duplication is linked to anxiety, but these results have been challenged by independent studies showing that GLO1 duplication has no effect on anxiety [20]. It is clear that GLO1 plays a key role in MGO removal; however, GLO1 overexpression can inhibit MGO accumulation, while GLO1 inhibition leads to MGO clustering [19,21]. GLO1 overexpression augments anxiety-like behaviour across different genetic backgrounds and in multiple behavioural tests [22,23,24,25]. Taken together, these results further underscore the importance of GLO1 based on its impact on anxiety-associated behaviour through the neural circuit, which is typically associated with anxiety-like behaviours. Remarkably, numerous studies have identified a relationship between GLO1 and behaviour. GLO1 expression may be associated with anxiety-like behaviour [19,61,62], depression [63], and neuropathic pain [64,65]. Oxidative stress [65] and sleep-deprivation models of psychological stress in phenotypically anxious rats [19] have been established by observing the role of GLO In addition, habit, motor activity, and motor harmony have been shown to be linked to GLO.

Excessive activation of Glo1 in vitro prevented MGO accumulation under high-glucose circumstances [66], and in *C. elegans*, a large secretion of Glo1 reduced basal MGO concentration [55].

The excess secretion of Glo1 has been shown to reduce cerebral MGO in mice [19]. Taking into account the strong correlation between Glo1 and MGO concentration, Glo1 could directly affect behaviour by controlling MGO levels. Recently, the behavioural effects of MGO, including the effects related to anxiety-like behaviour (63), depression [67], motor coordination [19], and pain [65], have been studied. Together, MGO concentration coregulates a variety of behavioural phenotypes, and the effect of MGO is opposite that of Glo1 overexpression. Some behavioural tests have used crude MGO preparations, such as pollutants such as formaldehyde and methanol. In particular, in these tests, it was found that laminin and fibronectin were modified by MGO [68]. The levels of AGEs increased in the brains of mice that were given long-term MGO treatments. The activation of the GABAA receptor was also found to be a strong alternative mechanism for MGO’s antianxiety effect.

### 3.5. MGO and Neuropathic Pain 

Neuropathic pain is a common sequela of diabetes and one of the features of neuropathy. Studies have shown the connection between Glo1 and neuropathy (18). However, it is crucial to confirm the relationship between the Glo1 system and behaviour during neuropathic pain. Here, it has been shown that A/J mice carrying the triple Glo1 allele can tolerate higher levels of mechanical pain than diabetic B6 mice, which only have a single Glo1 copy. Consistent with previous results, Glo1 was found to be enhanced in BALB/cByJ mice to protect against diabetic hyperalgesia when compared with BALB/cJ mice [19]. In addition, a recent study confirmed Glo1’s protective effect on diabetic neuropathic pain. The upregulation of human Glo1 decreased thermal hyperalgesia in diabetic mice [65]. Together, these results show that Glo1 plays an essential role in regulating behavioural measures of pain, especially behavioural measures associated with diabetic neuropathy. 

### 3.6. Other Mental Disorders and the MGO-Glo System

Human studies have indicated that the coding of single-nucleotide polymorphisms (SNPs) in GLO1 is associated with autism. However, because of a small sample size and potential confounding variables, the significance of this study is limited. For Finnish patient participants, there was no significant relationship or association between SNPs in GLO1 and autism [69]. Furthermore, based on a family-based clinical study, the 419A allele is more commonly found in unaffected siblings of autistic patients, and the difference in its enzymatic activity is attributable to the 419A GLO1 allele. In addition, in post-mortem brain samples from autistic patients, GLO1 activity was diminished, and AGE levels were higher than those of patients in the control group. Nonetheless, some human-genome-related studies (GWAS) have been unable to determine the correlation between GLO1 polymorphism and autism, and further studies should explore the functional significance. The presence of 410 A/C SNP and its effect on autism in mice can show whether GLO1 may be a contributing factor to autism. It is recommended to use a variety of behavioural tests to measure autistic phenotypes, including a social interaction evaluation tool, repetition assessment, a stereotyping behaviour estimate, and backward learning curves [70].

Two relevant studies have recently shown that GLO1 plays an essential role in schizophrenia. In a single schizophrenic patient, a frameshift mutation in GLO1 was shown to be correlated with diminished GLO1 enzymatic activity [71]. A previous study reported that the release of AGE aggregates is elevated in schizophrenic patients compared with members of the control group [72], indicating a diminished GLO1 function in schizophrenic patients. Research on human genetics and GLO1 in mice can contribute to establishing a link between GLO1 function and schizophrenia. Behaviour analyses of schizophrenic behaviour should be performed to research exceptional GLO1 expression.

### 3.7. MGO and Atherosclerosis, Hypertension, Ageing, and Epigenetics

Remarkably, the brain’s protection from glycation, much like with GSH and GLO1, declines with ageing. This indicates that MGO levels can become elevated with ageing and in patients with PD. Therefore, identifying the molecular mechanisms by which glycation modifies protein homeostasis and how it contributes to α-synucleinopathies would be a vital breakthrough in establishing the link between ageing and neurodegenerative disorders as well as a novel therapeutic strategy for α-synucleinopathies. In addition, dicarbonyl stress may be a mediator of obesity and insulin resistance. The association between dicarbonyl stress and senescence has been established in the functional genomics of GLO1 in the nematode *C*. *elegans* [54]. MGO-derived dicarbonyl stress may be related to plant ageing. MGO-H1 content was highest in Arabidopsis thaliana leaves, and the content of dicarbonyl in broccoli increased with age. After MGO exposure, the levels of aldosterone, renin, angiotensin, and phentolamine were significantly elevated after methyl acetaldehyde treatment. There were significant increases in MGO, angiotensin protein and mRNA, AT1 receptor, adrenergic α1D receptor, and renin. 

Furthermore, MGO stimulates NF-κB through RAGE and elevates renin and angiotensin levels, a new finding that may reveal one of the mechanisms underpinning elevated blood pressure. Here, GLO1 system and MGO levels were elevated in patients with atherosclerosis and diabetes. Recent studies have shown that GLO1 and MGO can lead to dysfunction of the vascular endothelial function [73]. Glycated end and metabolic oxidative products are implicated in severely protracted subclinical atherosclerosis, which plays a vital role in the negative metabolic memory of major vascular complications in humans [73]. MGO-induced oxidative stress—and not CpG demethylation—can epigenetically and rapidly inhibit the expression of the sFRP-4 gene, indicating protracted oxidative stress, such as in diabetes mellitus and ageing. It has been reported that MGO rapidly glycates proteins, damages mitochondria, and induces a prooxidant state similar to that observed in aged cells [19]. Indeed, AGE-modified proteins are plausibly involved in the pathology and etiopathogenesis of ageing.

## 4. MGO Signalling Pathway: Role and Mechanism in Seizures

MGO is a newly recognised type of epileptic seizure inhibitor. Epilepsy is a neurological disorder characterised by susceptibility to seizures. In a study, mice were pre-treated with MGO before convulsion induction, and picrotoxin or pilocarpine was used to assess seizures behaviourally or via electroencephalography (EEG). The researchers investigated the causal relationship between GLO1 and the assessed behaviour and the electroencephalogram readings induced by MGO detoxification during epileptic seizures in transgenic mice with overexpressed GLO In animal epilepsy models, a decrease in GLO1 activity leads to inadequate elimination of MGO, which can alleviate epileptic seizures. Nonetheless, MGO concentration can be overloaded when accompanied by AGE aggregation, which can also stimulate RAGE, leading to an inflammation cascade and a reduction in epileptic seizure episodes [74].

MGO is an endogenous regulator of seizures. Meanwhile, GLO1 suppression reduces seizures, indicating that this may be a novel therapeutic approach for epilepsy. In line with these characteristics, GLO1 may be associated with the genetic structure of epilepsy because GLO1 expression regulates MGO concentration levels and the severity of seizures [75].

Furthermore, considerable evidence suggests that mutations in GABAA receptor coding genes interfere with GABAA receptor signalling, triggering epileptic seizures [75]. Furthermore, some mature AEDs activate or enhance GABAA receptors, including benzodiazepines and barbiturates [76]. Differences in GLO1 expression and activity can influence epilepsy susceptibility via modulation of the concentration of endogenous MGO in the brain.

In addition, MGO exposure has been shown to decrease picrotoxin-associated and pilocarpine-associated seizures. The effect of MGO in two epilepsy seizure models demonstrate the extensive antiepileptic function of MGO. These findings are consistent with the role of MGO as a GABAA receptor agonist in the CNS system, which mediates neuronal inhibitory tension [75]. Three important indicators of the effect of MGO on epileptic seizures are the incubation period, seizure duration, and duration of the first seizure. In general, at first glance, MGO, an endogenous GABAA receptor agonist, can prevent seizures. Specifically, pre-treatment with MGO decreased the proportion of mice exhibiting convulsive behaviour during picrotoxin-dependent seizures. However, there is promise in further studying the therapeutic effects of MGO for epilepsy treatment. 

## 5. Potential Role of Small Molecules Such as MGO and GLO1 in Neuroprotection and Relevant Diseases

Remarkably, these cells, even in normal environments, had a higher proliferation rate than all other cells, which may demonstrate that overexpression of GLO1 may have a universally positive effect on viability. A GLO1 knockdown can significantly reduce cell proliferation and migration. Here, the protective effect of venlafaxine on vascular endothelial cells was observed in several models.

When induced by MGO in the microvascular endothelial cells of the human brain, venlafaxine had a protective effect on cell damage and significantly reduced the level of reactive chemical species. In the same study, caspase-3 and BAX proteins were diminished, which elevated BDNF and Bcl-2 protein release in the cultured microvascular endothelial cells of the human brain. Furthermore, venlafaxine is restrained from MGO-induced JNK phosphorylation. Venlafaxine was shown to enhance AKT phosphorylation, and the PI3K/AKT inhibitor restrained from venlafaxine, demonstrating that venlafaxine had a protective effect on MGO-induced microvascular endothelial cell damage via the PI3K/AKT or JNK pathway, which may be the underlying mechanism of microvascular endothelial cell damage [77]. 

Taking GSH as a catalyst and GLO1 and GLO2 as sequential actors, MGO was shown to decompose into D-lactate [78]. GSH is consumed by oxygen free radicals generated after ischaemia and cerebral ischaemia, which leads to the failure of the GLO system, and here, the enhancement of glutathione intervention may promote the GLO system by catalysing catechol catabolism [79]. 

Because pyruvic acid salt is highly soluble in an aqueous solution, there is the question of whether pyruvate can be used as a quick intervention measure to protect the brain from glycosylation stress caused by MGO. The required high concentration of pyruvic acid salt solute is easy to prepare, and it is very clear that the system in a loop to achieve effective treatment of pyruvic acid concentration needed for the infusion volume is minimised. Pyruvate passes through the blood–brain barrier of cerebrovascular endothelial cells via monocarboxylic acid transporters [80,81], delivering the compound to brain parenchyma.

Ischaemia reperfusion caused by cardiac arrest (CA) and cardiocerebral resuscitation (CCR) can lead to oxidative and carbonyl stress, damaging the brain. A shift from ischaemia to anaerobic glycolysis, along with the complication of oxyradical inactivation of GAPDH, can cause overproduction of MGO, which is a powerful glycating agent. In a study using CA and CCR, swine models demonstrated that intense cerebral ischaemia reperfusion induced by CA resuscitation can disable GLO1 and glutathione reductase (GR). Furthermore, the administration of pyruvate can hinder the inactivation of GLO1 and GR and stem protein glycation in the cerebral cortex. These findings have shown that GLO inactivation can result in cerebral protein glycation, suggesting that the use of pyruvate can help stop protein glycation in a postischemic brain [78]. 

After MGO exposure, a reduction in Trx1 and GLO2 can stem from autophagy inhibition. AMPK activation is significantly increased by MGO treatment. It has been shown that in AMPK-deficient cells, Trx1 and GLO2 in wild-type mouse embryonic fibroblasts are diminished. However, MGO has been shown to have no effect on these two substances. Generally, MGO stimulated autophagy in an AMPK-dependent mode, and autophagy may underpin Trx1 and GLO2 degradation, which would prove that Trx1 and GLO2 are the targets of MGO in HT22 nerve cells. 

A previous study indicated that diabetes can aggravate the amount of brain damage associated with stroke, which is related to the state of MGO-to-GSH in the brain. Brain damage was reversed by NAC. The prethrombotic phenotype in systemic circulation and brain tissue during diabetes was shown to be linked to the elevated MGO-glycation of proteins, which may be inhibited by NAC. Therefore, the diabetic brain and blood gradually become more impressionable to platelet activation and thrombosis. NAC can alter systemic and vascular prethrombotic reactions by increasing platelet GSH and GSH-dependent MGO removal after diabetes presents, which may protect against the risk of stroke and amend the levels of antioxidants in the body, including SOD1 and GPx-1 [80]. Dietary components not only remove MGO but can also affect certain biochemical events along a signalling pathway (signal transduction, stress protein synthesis, and nonenzymatic glycosylation) related to PD pathology. A study of double-blinded, placebo-controlled patients showed that supplementing myosin has beneficial effects for the body. MGO cleaners such as carnosine should be studied in more depth to explore their therapeutic strategies towards PD. Notwithstanding these findings, other compounds and drugs have shown promise, both experimentally and clinically, in limiting complications from diabetes. Pyridoxamine has shown some advantages in relevant trials of diabetic nephropathy, including a decrease in urinary N(epsilon)-(carboxymethyl)lysine and N(epsilon)-(carboxyethyl)lysine [82]. Metformin is a drug commonly used to treat type 2 diabetes, and it can decrease reactive dicarbonyls and AGEs [83,84]. Some additional compounds, such as aspirin, pioglitazone, benfotiamine, angiotensin-converting enzyme inhibitors, angiotensin II receptor blockers, and thiamine, have an anti-AGE function [85]. After long-term use, soluble RAGE (sRAGE), a decoy receptor for AGEs, shows promising results because of its ability to stall the development of sensory defects in diabetic mice [86].

Lewy body disease patients have been shown to have a similar staining pattern to patients with AGEs. Regulation of most parts of GLO1 can influence the accumulation and toxicity of α-synuclein. GLO1 is the main line of defence against MGO, a dicarbonyl metabolite inevitably formed through multiple catabolic processes [87]. GLO1 activity depends on diminished GSH, and MGO has been linked to GSH in the formation of MGO-GSH (a substrate of GLO1) and NADPH [88]. Consequently, MGO metabolism reduces the levels of these cofactors. Notably, when MGO is elevated, it contributes to higher glycation levels in patients with high blood sugar and patients with diabetes [89,90]. MGO concentration levels may rise with age and in patients with PD [90]. Therefore, deterministic glycation alters the relevant mechanisms of protein balance, leading to α-synucleinopathies, which may be a breakthrough in establishing the link between ageing and neurodegeneration, revealing a novel strategy for prevention.

The primary redox-sensitive signals and enzyme system play a key role in the adaptive cellular protective mechanism, which is an important stress response defence enzyme. The metabolism of MGO is relevant to the protection of cells from glycation and oxidative stress [91]. The Nrf2-HO-1 and Nrf2-GLO1 pathways convey adaptive antioxidant reactions that neutralise the function by controlling ROS and MGO aggregation. Nevertheless, the internal conditions for the reduction–oxidation reaction occur at the expense of another cellular protective mechanism.

GLO1, one of the major downstream targets of Nrf2 transcription activity, is a vital stress response defence protein that protects cells from dicarbonyl glycation and oxidative stress [91]. Cannabidiol, a non-psychoactive compound, has an antiepileptic effect and can be found in marijuana. The protective effect of cannabidiol on OGD/R damage depends partially on reducing oxidative stress. Promote mitochondrial bioenergy and regulate glucose catabolism through the pentose-phosphate pathway, thus maintaining the energy and balance of the reduction–oxidation reaction [92].

The GLO1 inhibitor, S-bromobenzylglutathione cyclopentyl diester (pBBG), has been used to lower EtOH consumption. GLO1 overexpression is on the contrary. Data suggest that EtOH reduction induced by GLO1 inhibition does not give credit to elevated EtOH stimulation or inhibition [93]. GLO1 has been shown to be linked to programmed cell death induced by Min-U-Sil-5 crystalline silica. Marein is a natural phenol for resisting diabetic encephalopathy and can reduce the p-AMPK-associated loss of activity while interacting with a subunit of AMPK in a specific orientation activation [94].

Casein may be an effective compound for preventing or combating diabetic encephalopathy, and casein preconditioning can reduce C-induced p-ampk inactivation. A molecular docking simulation showed that marein interacts with AMPK subunits. Here, autophagy is a probable mechanism via which the loss of GLO2 and Trx1 caused by MGO is induced. This process is activated by MGO in an AMPK-associated mode and is one of the factors responsible for the degradation of the molecular targets of MGO, such as Trx1 and GLO An aggregation of reaction metabolites occurs in diabetic retinopathy, including the ROS, RNS, and RCS species, which have been reported to regulate the activity of the transient receptor potential cation channels. Diabetic retinopathy is caused by the accumulation of reactive metabolites such as ROS, RNS, and RCS; it has been reported that RNS and RCS can regulate the activity of the transient receptor potential CAT family. GLO1 is the main detoxifying enzyme of MGO and is remarkably elevated in the Trpc1/4/5/6-deficient cell line, resulting in augmented resistance to MGO-induced damage. The expression of TRPCs in retinal endothelial and glial cells are at different levels. It has also been shown that a knockdown of GLO1 leads to apoptosis, the accumulation of MGO, and cytotoxicity [95]. Pyruvate kinase M2 activation may protect against diabetic nephropathy by increasing glucose metabolic flux, inhibiting the production of toxic glucose metabolites, and inducing mitochondrial biogenesis, thus restoring mitochondrial function [96]. Here, *Drosophila melanogaster* and *Caenorhabditis elegans* can be used to explore AGE-related pathways in depth, and to identify and assess the drugs that may mitigate the detrimental effects of the development of AGE adducts [97]. The deglycase DJ-1 (PARK7) protects histones from MGO adduction [98]. Notwithstanding recent advances in treatment, this disease requires further research, specifically regard to nosogenesis, as the intervention strategies for MGO and GLO1 presently focus on small nutrition molecules.

## 6. MGO and Homeostasis

MGO may be an essential molecule in the regulation of neural homeostasis (redox homeostasis, lipid metabolism homeostasis, energy homeostasis, protein steady-state, epigenetic mechanism, and neurotransmitters). The bioenergy and redox requirements of the brain are crucial for neural activity, with the astrocyte-neuronal lactate shuttle interpreting the energy requirements for nerve transmission. Furthermore, synaptic transmission inevitably promotes an increase in reactive chemical species in the mitochondria of neurons. Correcting impaired catalytic steps in glycolysis and mitochondrial bioenergetics may alleviate hyperglycaemia. By enhancing glycolysis and mitochondrial bioenergetics, glucose homeostasis can be dramatically improved. Constituted oxidant and proinflammatory conditions can induce chronic adaptive redox homeostasis, making cells more sensitive to additional stress events, especially considering that they are unable to reconstruct a state of steady internal conditions. In addition, functional incapacitation in some pathways contributes to constant changes that lead from normal homeostasis to stress reaction, forcing cells to adapt to chronic mild stress. Neurons are particularly susceptible to oxidative stress because they lack the capacity to prevent oxidation and maintain energy under steady internal conditions. The GLO1-MGO pathway integrates maternal and cell metabolism to regulate neural development, and perturbations in this pathway can lead to long-lasting alterations in adult neurons. GLO1, when mutated, is associated with neurodevelopmental disorders. The existence of direct inter-pathway communication between glycolysis and the KEAP1-NRF2 transcriptional axis can provide more insight into the metabolic regulation of the cellular stress response, suggesting a therapeutic strategy for controlling the cytoprotective antioxidant response. The mechanisms that integrate the metabolic state of a cell with regulatory pathways are necessary for maintaining cellular homeostasis [99].

Future research should aim to move beyond the notion that glycolysis is a source of protein homeostasis destruction. GLO1 levels are critical to MGO detoxification, which can prolong nematode life span, while its reduction can reduce life span. MGO promotes the formation of altered proteins and compromises their proteasomal elimination, which further promotes age-associated protein steady-state malfunction. A detailed elucidation of the epigenetic mechanism and molecular pathways through which MGO-induced neurotoxicity occurs will help.

## Figures and Tables

**Figure 1 molecules-27-07905-f001:**
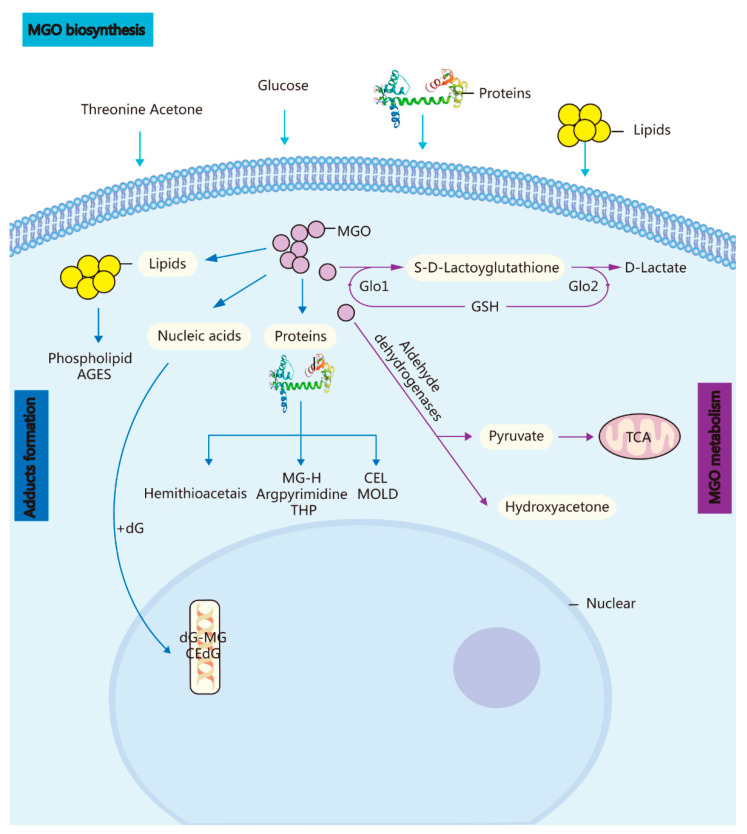
Biosynthesis, metabolism, and adduct formation in MGO. MGO biosynthesis: MGO is an unavoidable endogenous coproduct of the metabolism of glucose, threonine acetone, lipids, and proteins. Adduct formation: The incubation of these highly reactive compounds with proteins, lipids, and nucleic acids leads to the rapid formation of AGEs and advanced lipoxidation end-products. MGO metabolism: MGO is converted into S-D-lactoylglutathione in the GSH-dependent GLO1 system and subsequently transformed into D-lactate via GLO Methylglyoxal reductase and aldehyde dehydrogenase convert MGO into hydroxyacetone or pyruvate.

**Figure 2 molecules-27-07905-f002:**
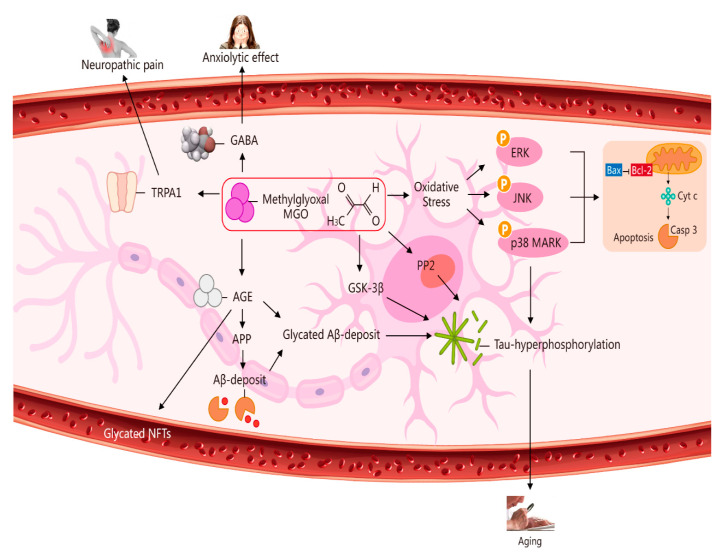
Multiple pathways via which MGO induces oxidative stress, pain, anxiolytic effects, and ageing. The pathways activated by MGO induce oxidative stress, which may play a role in ageing. The formation of AGEs involves nonenzymatic reactions that occur when sugars or dicarbonyl compounds, such as MGO, are reduced. Research has been conducted on the connection between neuropathic pain and the transient receptor potential ankyrin 1 (*TRPA1*) pathway. MGO mediates the anxiolytic effect via the modulation of GABAergic actions. MGO mediates ERK-, JNK-, and p38-dependent endothelial cell inflammatory responses, which may occur independently of oxidative stress. In addition, BCL2-associated X protein (BAX) accesses mitochondria to induce cytochrome *c* release, and Bcl-2 inhibits BAX translocation from cytosol to mitochondria during relevant apoptosis. MGO induces the expression of phosphorylated tau protein and increases the expression of protein phosphatase 2A (PP2A) and glycogen synthase kinase-3β (GSK-3β). A reduction in Aβ deposition can eventually improve learning and memory ability. AGE interaction with the receptor for AGEs (RAGE) carries an implication for AD. Glycated Aβ (Aβ-AGE) can aggravate an AD-like pathology via the receptor for the RAGE pathway. MGO increases intracellular calcium in sensory neurons and produces neuropathic pain via the cation TRPA1 channel.

**Figure 3 molecules-27-07905-f003:**
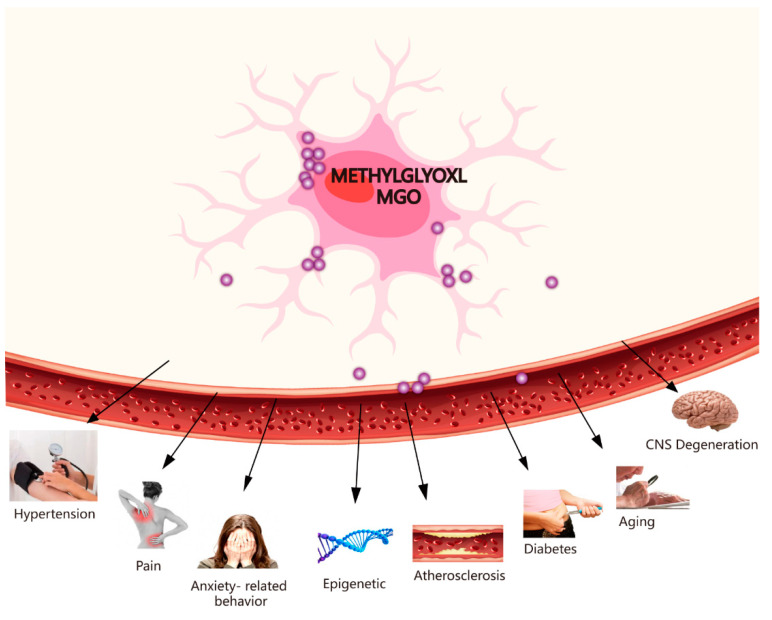
MGO-induced diseases. MGO-induced diseases involve hypertension, pain, anxiety-related behaviour, epigenetic, atherosclerosis, diabetes, ageing, and CNS degeneration.

## Data Availability

All data supporting our findings are included in the manuscript.

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
