# Peer review of "Methylglyoxal in the Brain: From Glycolytic Metabolite to Signalling Molecule"

_molecules, 2022, doi:10.3390/molecules27227905_

Round 1
Reviewer 1 Report
The review article "Methylglyoxal in the brain: from glycolytic metabolite to signalling molecule" by Yang et al. is a good topic. Addressing the following suggestions would improve the manuscript presentation.
Major concerns:
Figure 1 title is Biosynthesis, metabolism and adducts formation. The biosynthesis of MGO is not shown in figure 1. The figure shows that neurons have abundant MGO, leading to many diseases. Describe it clearly.
Figure 2 shows the biosynthesis of MGO from various sources, a by-product of the MGO, and its connection to the metabolism have shown clearly. Still, the relevant diseases are not shown in the figure, but it was mentioned in the figure title.
The third figure describes the MGO connection with neurological and neurodegenerative diseases but not to diabetics or epigenetics. The figure needs to be vividly explained in the legend.
Minor concerns:
Page1, paragraph 2, line 2 is not very clear. Mention the homeostasis name.
Page3, paragraph 2, line 1: Mention the MGO involved neurodegenerative diseases names. Correct the irregular font sizes in a few words. The last paragraph of the first line needs to give a full meaning. Complete the sentence where MGO induces apoptosis and mention the citation.
Page8, 3.2, lines 3-6 are repetitive.
Mention the C.elegans name uniformly and correct all irregular font sizes throughout the review.
Author Response
Please the file named Response to Reviewer 1 point to point.docx

Reviewer 2 Report
1. The author may provide drug-related information regarding MGO side effects.
2. An author should include current research or target regarding MGO.
3. An author should discuss more future research on MGO.
4. The author shall correct some typographical errors in the manuscript
Author Response
Please see the file named Response to Reviewer 2 point to point

Round 2
Reviewer 1 Report
Dear Authors,
The revised manuscript has s improved a lot and gives a good flow to the readers.